# Machine Learning Technology Reveals the Concealed Interactions of Phytohormones on Medicinal Plant In Vitro Organogenesis

**DOI:** 10.3390/biom10050746

**Published:** 2020-05-11

**Authors:** Pascual García-Pérez, Eva Lozano-Milo, Mariana Landín, Pedro Pablo Gallego

**Affiliations:** 1Applied Plant & Soil Biology, Plant Biology and Soil Science Department, Biology Faculty, University of Vigo, E-36310 Vigo, Spain; pasgarcia@uvigo.es (P.G.-P.); e.lozanomilo@gmail.com (E.L.-M.); 2CITACA—Agri-Food Research and Transfer Cluster, University of Vigo, E-32004 Ourense, Spain; 3Pharmacology, Pharmacy and Pharmaceutical Technology Department, Faculty of Pharmacy, University of Santiago, E-15782 Santiago de Compostela, Spain; m.landin@usc.es; 4Instituto de Investigación Sanitaria de Santiago (IDIS), E-15782 Santiago de Compostela, Spain

**Keywords:** algorithms, artificial intelligence, auxins, cytokinins, in vitro culture, *Kalanchoe*, plant growth regulators (PGRs), plant tissue culture

## Abstract

Organogenesis constitutes the biological feature driving plant in vitro regeneration, in which the role of plant hormones is crucial. The use of machine learning (ML) technology stands out as a novel approach to characterize the combined role of two phytohormones, the auxin indoleacetic acid (IAA) and the cytokinin 6-benzylaminopurine (BAP), on the in vitro organogenesis of unexploited medicinal plants from the *Bryophyllum* subgenus. The predictive model generated by neurofuzzy logic, a combination of artificial neural networks (ANNs) and fuzzy logic algorithms, was able to reveal the critical factors affecting such multifactorial process over the experimental dataset collected. The rules obtained along with the model allowed to decipher that BAP had a pleiotropic effect on the *Bryophyllum* spp., as it caused different organogenetic responses depending on its concentration and the genotype, including direct and indirect shoot organogenesis and callus formation. On the contrary, IAA showed an inhibiting role, restricted to indirect shoot regeneration. In this work, neurofuzzy logic emerged as a cutting-edge method to characterize the mechanism of action of two phytohormones, leading to the optimization of plant tissue culture protocols with high large-scale biotechnological applicability.

## 1. Introduction

Recent reports have highlighted that the 25% of all drugs approved by the Food and Drug Administration (FDA) proceed from plant sources [1]. From an industrial point of view, plant in vitro tissue culture constitutes a successful technology for large-scale processes: it offers an enhanced yield stability and quality of plant by-products and, at the same time, it enables the inclusion of different applications under controlled conditions to respond to industrial requirements [2]. In this sense, a plethora of strategies has already been applied to that end, such as metabolic engineering, elicitation and culture media optimization [3]. Thus, plant in vitro regeneration constitutes one of the basic strategies that are commonly applied in plant biotechnology for the exploitation of medicinal species and it is driven, at a cellular level, by plant in vitro organogenesis. Organogenesis is a highly complex feature that takes advantage of plant cell totipotency to form new organs with the ability of developing into functional plantlets, under specific conditions [4]. The complexity associated with the understanding of plant organogenesis is mostly due to a series of factors that drives this process, as it depends on both genetic and environmental factors. Concerning genetical factors, the molecular knowledge behind organogenesis has been focused on a high number of studies performed on *Arabidopsis* [5]. Although such studies shed light about the molecular basis of this process, additional works concluded that organogenesis is highly dependent on the genotype studied, and universal assumptions should be avoided [6]. Environmental factors also play a key role in organogenesis; however, under in vitro conditions, most of these factors are represented by culture media components and growth conditions, thus emerging as a convenient tool for developing experimental studies in the field of organogenesis [7]. Among the culture media constituents, natural plant hormones, known as phytohormones, act as plant growth regulators (PGRs) that modulate the occurrence of signaling events during organogenesis. More precisely, auxins (AXs) and cytokinins (CKs) are considered the principal phytohormones involved in this process [8]. They both work coordinately to guide different organogenetic responses towards the formation of new structures that give rise to organs, through a complex intracellular crosstalk that has not been fully elucidated to date [9].

Common protocols for in vitro organogenesis and plant regeneration are based on the transference of explants onto culture media containing different exogenous PGRs, including both natural and/or synthetic hormones. Such molecules then cause a regenerative process driven by the development of adventitious shoots, which can eventually undergo rooting to form fully developed plantlets. Nevertheless, this process may occur either directly, from explant cuttings to shoots, or indirectly, via callus formation [10]. On the other hand, one of the factors that should be considered when developing experimental protocols based on organogenesis, is the induction of somaclonal variation, which may induce relevant modifications that can interfere with the genetic fidelity of plant multiplication [11]. In general, genetic instability is inherent to plant tissue culture; however, the method involved in multiplication determines the incidence of somaclonal variation. In this sense, direct shoot regeneration minimizes the chance of genetic instability, compared to callus-mediated regeneration [12], so the role of exogenous phytohormones goes beyond phenotypical traits and, consequently, it is mandatory to achieve an adequate phytohormonal balance for each genotype in order to accomplish efficient plant tissue culture protocols [13].

Due to their multifactorial behavior, the design of universal protocols for plant regeneration is still a challenging task: the impossibility of predicting the wide range of biological events that takes place during organogenesis, forces the development of specific, complex empirical studies applied to different genotypes. Such complexity is mostly due to the large amount of experimental data and their heterogeneous nature, which motivate the construction of unmanageable databases that make the analysis and interpretation of results using traditional statistical methods difficult [14]. As a solution, the use of machine learning (ML) approaches has been recently proposed [15]. Machine learning algorithms have the ability of predicting and characterizing complex processes, which include multiple variables, thus becoming an efficient, predictive decision-making tool [16].

ML technology relies on the combination of computational statistics with algorithms [16], thus resulting in robust mathematical models constructed from a dataset that includes several independent variables or factors (named as inputs) and dependent variables or responses (named as outputs). Hence, the use of different ML approaches, such as artificial neural networks (ANNs), stands out as a novel and efficient approach to provide insight about the underlying multifactorial mechanisms driving plant organogenesis. The combination of ANNs with fuzzy logic, known as neurofuzzy logic, becomes a powerful tool for data modeling, thus making the optimization and prediction of complex processes easier and, at the same time, it provides a simplistic interpretation of results, by the generation of “IF–THEN” rules [17]. Furthermore, this computer-based tool is able to range both input and output values at different levels (low, mid, high, etc.) according to the generated model, combined with a corresponding membership degree value that varies between 0 and 1 [14].

The use of this artificial intelligence-based tool has been successfully applied in the field of plant tissue culture, as it was demonstrated for germination [18], micropropagation [19] and the enhancement of phenolic compounds production [15].

In this work, we will take advantage of the use of algorithms, instead of traditional statistics, for deciphering the role and concealed interactions existing between exogenously applied phytohormones, AXs and CKs, on the organogenesis of unexploited medicinal plants belonging to the *Bryophyllum* subgenus. *Bryophyllum* spp. constitutes a large group of plants from the genus *Kalanchoe* (Crassulaceae), commonly known for their uses in traditional medicine, across Africa and Asia, for the treatment of chronic disorders, including diabetes and neurodegenerative, cardiovascular and neoplastic diseases [20]. To date, limited information is available about the in vitro propagation of *Bryophyllum* species. Consequently, our study is committed to provide insight into the critical factors that influence the in vitro organogenesis of *Bryophyllum* spp., by focusing on the effects developed by the exogenous application of two phytohormones: the auxin indoleacetic acid (IAA) and the cytokinin 6-benzylaminopurine (BAP).

## 2. Materials and Methods

### 2.1. Plant Material

Three different species belonging to the *Bryophyllum* genus were subjected to the establishment of plant in vitro culture: *Bryophyllum daigremontianum* Raym.—Hamet et Perr. (BD), *Bryophyllum × houghtonii* D.B. Ward (*Bryophyllum daigremontianum × tubiflorum*, BH) and *Bryophyllum tubiflorum* Harv (BT). Epiphyllous plantlets from adult plants grown in a local greenhouse (42°12′40.0′′ N 8°43′36.1′′ W) were collected and disinfected according to previous works [21]. Plantlets were later transferred by groups of three individuals to culture vessels containing 25 mL of previously autoclaved Murashige and Skoog (MS) medium [22], supplemented with 3% (*w*/*v*) sucrose and solidified with 0.8% (*w*/*v*) agar at pH = 5.8. Cultures were placed into a growth chamber with a photoperiod of 16 h light (55 µmol m^−2^ s^−1^) and 8 h dark at 25 ± 1 °C. Plantlets were subcultured every 12 weeks to fresh media, by using newly-formed epiphyllous buds as the new propagation explants, and plants from the 6th subculture (after 72 weeks) were used here as the source of foliar disks for in vitro organogenesis experiments.

### 2.2. Organogenesis Experiments

Foliar disks (≅1 cm^2^) were excised and used for the subsequent in vitro organogenesis experiments. The leaf disks (3) were placed into each culture vessel (4) and used for each treatment, making up a total of 12 biological replicates (*n* = 12). All vessels were placed randomly in a growth chamber under the same conditions described above.

The culture medium for such experiments consisted in the previously described MS medium supplemented with different concentrations of two phytohormones: the auxin indoleacetic acid (IAA) and the cytokinin 6-benzylaminopurine (BAP). More concisely, four different levels of each phytohormone were used: 0, 0.25, 0.5 and 1 mg L^−1^, which constituted 16 treatments (Table 1).

After 8 weeks, six parameters were determined to analyze in vitro organogenesis from the responses observed on the foliar disks:Percentage of direct shoot regeneration (%DS): percentage of explants showing newly formed direct shoots.Percentage of indirect shoot regeneration (%IS): percentage of explants showing newly formed indirect shoots.Number of direct shoots (NDS): number of newly formed direct shoots per explant.Number of indirect shoots (NIS): number of newly formed indirect shoots per explant.Percentage of callus formation (%CAL): percentage of explants showing callus formation.Percentage of direct rooting (%DR): percentage of explants showing direct rooting.

### 2.3. Modelling Tools

The combination of two ML approaches—neural networks (ANNs) and fuzzy logic, named neurofuzzy logic—was used to build the mathematical models [14,15]. Once all experimental data were collected, they were incorporated into one database and subjected to artificial intelligence analysis by FormRules^®^ 4.03 neurofuzzy logic software (Intelligensys, Ltd., North Yorkshire, United Kingdom). As stated earlier, the experimental design included a total of 48 combinations (3 × 2^4^), as a result of merging 3 different genotypes (BD, BH and BT) × 2 different phytohormones (IAA and BAP) × 4 levels and their interactions. Thus, a database was constructed including the 3 factors (selected as inputs): genotype, IAA and BAP concentrations; and the six organogenesis-related parameters indicated above (classified as outputs): %DS, %IS, NDS, NIS, %CAL and %DR.

For the model construction, the training parameters found in Table 2 were used. Cross validation (CV) was the algorithm used for model selection criteria. It drives the division of data into several subsets and one of them is taken out from the training set and later used for testing the results obtained by the training of the other subsets. As a consequence, the generated model offers a higher robustness of predicted values, by avoiding data redundancy [23]. Following model generation, the corresponding submodels were established by FormRules^®^ throughout the adaptative-spline-modeling-of-data mode (ASMOD).

One of the most important advantages in the use of neurofuzzy logic is the simplification of the results, by making their interpretation easier, as the results for inputs were expressed as “IF–THEN” rules. At the same time, the software was able to range both input and output values at different levels (low, mid, high, etc.) according to the generated model, combined with a corresponding membership degree value that varies between 0 and 1 [14]. The degree of membership represents a degree of truth, ranging from 0 to 1, meaning 1 that the expected output value is always a complete member of the fuzzy set “low”, “medium” or “high” [14,17].

In addition, FormRules^®^ provided independent predictive models for each output and their quality was evaluated throughout the determination coefficient of the training set, called Train Set R^2^, defined by Equation (1):(1)$$R^{2}=\left(1-\frac{\Sigma_{i=1}^{n}{\left(y_{i}-y_{i}^{\prime }\right)}^{2}}{\Sigma_{i=1}^{n}{\left(y_{i}-y_{i}^{\prime \prime }\right)}^{2}}\right)\times 100$$
where $y_{i}$ refers to the experimental values from the dataset, $y_{i}^{\prime }$ refers to the predicted values generated by the model and $y_{i}^{\prime \prime }$ refers to the mean of the dependent variable. The Train Set R^2^ was expressed as a percentage and significant predictive values ranged between 70% and 99.9%. Values above 99.9% were rejected as they are indicators of model overfitting [24]. Finally, analysis of variance (ANOVA) was performed in order to determine model accuracy, by evaluating the significant differences between the experimental data and predicted values obtained after fuzzification.

### 2.4. Statistical Analysis

Binary data (regeneration/no regeneration; callus/no callus; rooting/no rooting) reported for %DS, %IS, %CAL and %DR were statistically evaluated by binary logistic regression. Differences between treatments were analyzed by multiple comparisons adjusted by the Sidak method (*p* < 0.01), as suggested by other authors [25]. For count data derived from NDS and NIS, a non-parametric Kruskal–Wallis test was performed (*p* < 0.01), according to previous works [26]. The software used was SPSS 25 (IBM Corp., 2017, Armonk, NY, USA).

## 3. Results

The results from the organogenesis experiment are shown in Table 3. As it can be seen, the database constructed by the experimental data is large and the information derived from it is limited.

According to Table 3, most treatments (66.7%) induced direct shoot regeneration and the results for %DS showed that the maximum value (100%) corresponded to the genotype BH cultured on the treatment T04, which included the maximum concentration of BAP (1 mg L^−1^) in the absence of IAA. In the case of BD, the maximum value, 83.3%, was obtained with treatment T03 that exclusively included 0.5 mg L^−1^ BAP. In contrast, BT behaved differently, as the highest %DS value was significantly lower (*p* < 0.001) than the other species, 58.3% in the treatment T10, presenting a combination of 0.5 mg L^−1^ IAA and 0.25 mg L^−1^ BAP, with a lower efficiency.

The results for the percentage of indirect shoot regeneration (%IS) showed a very differential pattern with respect to %DS: (i) only 25% of the treatments promoted indirect shoot regeneration; and (ii) the maximum value was recorded for BT cultured on T03: 88.9% (Table 3). In contrast, for the rest of the genotypes, BH did not show indirect shoot organogenesis for any of the treatments used, whereas BD did, with a residual value of 8.3%, only on treatment T16, which contained the maximal concentration of both IAA and BAP. The comparison between direct and indirect shoot regeneration indicated that each process was individually caused by different factors and BT showed a differential behavior, which contrasts with the results obtained for BD and BH. These findings suggest the existence of underlying interactions between two or more factors that regulate both processes in every species.

The maximum number of direct shoots (NDS) was reported for BH on T08, which accounted for 3.7 direct shoots per explant (Table 3). For BD, the highest value was 2.5 direct shoots per explant, obtained on T15. Thus, BT showed the maximum NDS of 1.7 on the treatment T02. These observations show that NDS is independent of the rate of direct shoot regeneration (%DS), as they present different phytohormonal requirements, according to our results. The same observations were valid for the number of indirect shoots, NIS, where BT achieved a maximum value of 3.6 indirect shoots per explant with the treatment T14 (Table 3) that was supplied with a different combination of phytohormones than T03, which provide the maximum value observed for %IS.

Concerning the rate of callus formation, %CAL, results are related to those of %IS, since BH explants did not experience this process on any treatment tested and BD did only in two of them (T15 and T16). In contrast, all BT explants (100.0%) were able to develop calli on diverse treatments: T04, T06, T07 and T12 (Table 3).

Finally, in the case of percentage of direct rooting (%DR), almost none (85.4%) of the treatments induced direct rooting, as very low values were obtained for all species (Table 3) without statistical differences among the treatments (*p* < 0.001). The maximum values also were very low, in the range of 11–20%, for each genotype: BD (T08), BH (T13) and BT (T10). It should be noted that indirect rooting was not detected in this study and it was not further considered. Once again, each genotype showed different phytohormonal requirements for the induction of root formation, as it happened for shoot regeneration.

In summary, all these results demonstrated that simple statistical analysis gave much results about differences among the treatments but poor information (knowledge) about the key factors that control organogenesis, due to the non-linear and multifactorial interactions between the factors. The high complexity of the physiological phenomena that takes place during in vitro organogenesis, and the intricate interactions observed between phytohormones and genotypes, enabled the application of ANNs, combined with fuzzy logic, to reveal the relative significance of each factor and their hidden interactions on every process monitored during *Bryophyllum* organogenesis. Table 4 summarizes the results for the model generated by neurofuzzy logic.

The efficacy of neurofuzzy logic was demonstrated by the ability of predicting five out of six outputs, given by Train Set R^2^ values higher than 70% (Table 4). The only output that did not reach a significant predictability was %DR, whereas all the outputs related to shoot organogenesis and callus formation were efficiently predicted. The most important factors spotted by the model were the same for the outputs: %DS, %IS, NDS, NIS and %CAL (Table 4), providing key information on which factors caused the detected effects, specifically the interaction between genotype and BAP concentration. Additionally, for NIS, a second submodel was generated, with a lighter effect, concerning IAA concentration. The predictability of the model was assessed by the ANOVA ***f*** ratio, being always higher than the *f* critical values, highlighting that no statistical differences were observed (*p* < 0.05) between the experimental and the predicted values generated by the model (Table 4).

Although these results could seem simplistic, FormRules^®^ has the ability of generating the corresponding rules that define the influence of each factor on every output, as shown in Table 5. As it could be noted for each output, the model arranged the phytohormone concentrations into different levels, from “low” to “high” (Table 5), which were additionally defined by ML technology (Appendix A).

The neurofuzzy model predicted that culture media supplemented with low BAP concentrations (<0.125 mg L^−1^) always caused low %DS, independently of the genotype studied (membership degrees 0.98–1.00, Rules 1, 5 and 9; Table 5). On the contrary, the model suggests than other factors were involved in direct regeneration, because the %DS highest value was predicted at mid-high BAP concentrations (0.375–0.75 mg L^−1^) for BD (membership 0.71; Rule 3) and high BAP concentrations (>0.75 mg L^−1^) for BH (membership 0.85; Rule 8). It should be noted that, in all cases, %DS was predicted as low for BT, with independence of the BAP concentration (Rules 9–12).

In the case of %IS, the model exclusively revealed high predicted values for BT at mid BAP concentrations (0.25–0.75 mg L^−1^, Rule 20), the other values being low, with the strongest effects awarded to BD and BH independently of the phytohormone concentration (Rules 13–18).

In parallel, the results for NDS and NIS were in line with those of %DS and %IS, respectively (Table 5). NDS was high for both the BD and BH genotypes at high BAP concentrations, the latter showing the strongest effect (0.5–1 mg L^−1^, Rules 23 and 25), whereas BT showed low values for NDS in all cases (Rules 26 and 27). On the contrary, the only high value for NIS was predicted for BT at mid BAP concentrations (0.25–0.75 mg L^−1^, Rule 35), being low for the other two genotypes BD and BH, at any BAP concentration (Rules 28–33). In this case, IAA also played a significant role in NIS, as it was predicted as a negative factor on this output, since it caused low NIS values at any concentration (Rules 37–39).

In the same way, the prediction for %CAL was in line with the results related to indirect organogenesis, thus revealing that callus formation is crucial in such process: %CAL was high for BT, showing the strongest effect at mid and higher BAP concentrations (>0.375 mg L^−1^, Rules 49–51), whereas it was always low (membership 1.00) at low BAP concentrations (<0.125 mg L^−1^, Rule 48). Finally, for BD and BH, regardless of phytohormone concentrations, a low rate of callus formation was reported (membership 0.95–1.00; Rules 40–47).

For a better understanding of the influence of these two phytohormones on the organogenesis of *Bryophyllum* spp., the graphical interpretation for the prediction of each output is shown in Figure 1.

Remarkably, in addition to the rules, the 3-D graphical interpretation for the model indicates a clear influence of the genotype on the organogenesis of *Bryophyllum* spp., as it showed that BD and BH present similar organogenetic patterns (Figure 1A–J), which were clearly opposite to those found for BT (Figure 1K–O). Colors improve 3-D graphs interpretation. As an example, the rules described above for %DS (1–12; Table 5) can be visualized as if low BAP concentrations always promote low %DS responses (orange), and only high (dark blue) or mid-high (blue) concentrations drive high %DS values (Figure 1A,F,K). On the contrary, in the case of %IS, the 3-D graphs revealed high values (dark blue) for BT at mid BAP concentrations (0.25–0.75 mg L^−1^, Figure 1L), the rest of the values being low (green and orange zones). For BH and BD, %IS was always low, independently of phytohormone concentrations (red, Figure 1B,G). For the rest of parameters, a clear cause–effect relation can be observed on NDS (Figure 1C,H,M) and NIS (Figure 1D,I,N) for mid and high BAP concentrations for each genotype, following similar trends than %DS and %IS, respectively. In addition, the effect of IAA concentrations on NIS (Figure 1N) can be clearly observed, as described by Rules 37–39 (Table 5).

In general, the use of ML modelling facilitated the understanding of the results to reveal the concealed interactions of BAP and IAA on the development of organogenesis-related events on *Bryophyllum* spp. In order to show the power of ML technology in predicting and prioritizing the role and interactions of these two phytohormones, an integrative model was proposed in Figure 2 to characterize their effects on *Bryophyllum* organogenesis, focused on the major effects shown by BAP and genotypes.

## 4. Discussion

Medicinal plant research strongly depends on the establishment of efficient in vitro culture protocols to ensure a stable and scalable plant multiplication. For such purposes, organogenesis plays a critical role, as it drives the development of plant regeneration throughout the administration of different phytohormones. More specifically, AXs and CKs are considered the main PGRs used to that end, as their interactions guide the organogenetic process towards shoot regeneration, rooting or callus formation, among any other responses [27]. Thus, deciphering the role and effect of both phytohormones is a fundamental step that will determine the effectiveness of the plant regeneration protocols of the unexploited plants, as is the case of the medicinal species belonging to the *Bryophyllum* subgenus.

In order to accomplish such a goal, when designing plant in vitro protocols, the reliability and success on the production of true-to-type plants is essential for their industrial valorization and exploitation. In this sense, avoiding somaclonal variation is a crucial step when applying exogenous phytohormones. Two different phytohormones were tested in this study—the auxin IAA and the cytokinin BAP—at different concentrations, between 0 and 1 mg L^−1^. The preference to these PGRs responds to their natural origin, since they are found as natural phytohormones in plants. PGR origin is a usually underestimated factor in plant organogenesis, since the use of synthetic phytohormones normally increases the frequency of genetic variation, such as that of polyploidy [28]. In terms of concentration, high phytohormone concentrations may also provoke somaclonal variation-related phenomena, as is the case of polyploidy, caused by an excess of BAP, and an increased gene silencing, promoted by auxin excess, among others [29].

The results obtained for organogenesis allowed a primary approximation of this process on *Bryophyllum* spp. However, their interpretation became a really hard task, since values were spread into a large database and, moreover, the outputs contained a heterogeneous data nature, thus making their analysis difficult [15]. Taking a closer observation, some differential patterns could be analyzed, as it was the case of BT with respect to BD and BH, but the revealing of further information is limited, resulting in the impossibility of determining the role and interactions that take place between the different factors tested: genotype and BAP and IAA concentrations.

As a solution, the application of ML algorithms enabled the identification of significant factors that had an impact on almost every organogenetic parameter (Table 4). The only parameter that was not predictable after data modelling was %DR. However, this fact cannot be attributed to a lack of predicting power of this tool, as it can be explained on the basis of the explants used for this study: leaf-derived explants. As highly differentiated tissue, leaf explants should suffer an enormous cell reprogramming to give rise to differentiated roots via direct organogenesis and, eventually, higher auxin concentrations will be required to induce such a process [30]; other auxins, such as indole butyric acid (IBA), could be preferentially selected for that purpose [31]. Additionally, the presence of BAP inhibited root formation on BD, as reported by previous works [32].

Regarding all the other outputs, the model identified that the interaction between genotype and BAP concentration caused the most significant effect guiding *Bryophyllum* organogenesis (Table 4). Nevertheless, by taking a closer look at the corresponding rules (Table 5), BAP concentration played a major role in all the organogenetic variables, by dividing the different genotypes into two groups, BD and BH, which developed direct organogenesis-related events, as well as BT, which developed indirect organogenesis-related events (Figure 1). There is a large evidence base about the striking effect of genotype on plant organogenesis, even between closely related species, as it is in this case. Such an influence depends on the combination of both genotypical and phenotypical factors, which can be grouped into three major causes for application in this case:(1)The genetic control of differentiation-related genes, which includes a complex interaction between nuclear and cytoplasmic genes in the presence of PGRs, which may give rise to epigenetic changes during organogenesis [33].(2)The endogenous phytohormonal concentrations found in the starting explants, as they interact with exogenous PGRs used within the culture medium [34]. As a result, the balance between AXs and CKs determines the organogenetic response. Normally, the dominance of CKs over AXs, promotes cell division and shoot elongation; a balanced ratio between CKs and AXs promotes callus formation and embryogenesis; and AXs predominance results in root formation and elongation [35]. According to our results, the number of rules generated by the model (Table 5) pointed out the different BAP concentrations required to promote direct shoot regeneration (Rules 3 and 8), indirect shoot regeneration (Rule 20) and callus formation (Rule 50), depending simultaneously on the genotypes used. This indicates that, even if all genotypes were closely related, they could present a differential endogenous accumulation of phytohormones into leaves, as discussed later.(3)Leaf morphology: As succulent plants, *Bryophyllum* spp. accumulate high amounts of water within leaf tissues, including the three genotypes used in this work. However, BD and BH usually present a higher water content in comparison to BT, which presents a thicker cuticle layer that enables water accumulation to a lower extent [36]. This observation may be responsible for the differential pattern observed between direct and indirect organogenesis among these three species, since an increase in intracellular water accumulation is required during the initial steps of shoot regeneration [37]. Consequently, the higher water content found in the leaves of BD and BH may encompass the ability of these species to form direct shoots (Figure 1A–J). On the contrary, the indirect shoot regeneration observed for BT was guided by its high rate of callus formation (Figure 1K–O), which can be formed from cuticle-related tissues [38].

When leaf explants were cultivated in a phytohormone-free medium (0 mg L^−1^), no organogenetic response was observed, as leaf cuttings suffered oxidation from the first weeks of the experiment (data not shown). This phenomenon is related to the oxidative stress generated by explants after wounding and it was already studied in *Bryophyllum* plants cultured ex vitro and in vitro [15,39]. Moreover, the absence of phytohormones showed the same effect on BD, as previously reported [40].

In the presence of phytohormones, BAP was the main factor that drives organogenesis in all genotypes. Thus, low concentrations up to 0.375 mg L^−1^ BAP (Figure 2), led to subtle organogenetic responses without developing true usable shoots. Phytohormonal requirements, concerning mostly BAP, were also reported for BD by other authors, finding a minimum concentration of 0.6 mg L^−1^ BAP to observe organogenetic processes [32].

In the same way, intermediate BAP concentrations (0.375–0.75 mg L^−1^) showed different organogenetic patterns, depending on the genotype (Figure 2). Thus, under this range of BAP concentrations, BD and BH experienced a high frequency of direct shoot regeneration, whereas BT presented a high frequency of indirect shoot regeneration via callus formation, which was negatively influenced by IAA. The endogenous phytohormonal content could be responsible for this result, according to the foliar budding found in *Kalanchoe* species as an integral part of leaf ontogeny, which is the process in charge of their asexual reproduction [41]. This way, BD and BH may contain higher BAP endogenous concentrations, since they should maintain foliar budding along the leaf margins, while BT showed a restricted budding to the distal foliar apex (Figure 3), thus requiring a lower BAP content. This hypothesis is reinforced by the ability of BT to form callus without the addition of exogenous auxins, and it may indicate that this genotype probably contains a higher endogenous auxin content. Additionally, it would also explain the inhibiting effect on NIS developed by the addition of IAA, which may cause an auxin excess for indirect shoot regeneration (Figure 3). Such higher auxin concentrations found in BT was also demonstrated by other authors, since BT nodes were able to root spontaneously, without the addition of auxins [42,43].

In the case of high BAP concentrations (0.75–1 mg L^−1^, Figure 2), the same differential shoot regeneration behavior was observed, but NDS was higher for BD and BH, mostly due to the efficient role of BAP on shoot promotion and elongation. This finding was previously reported on BD, which developed the maximum NDS under 1 mg L^−1^ BAP [32]. On the contrary, BT experienced callus formation but indirect shoot regeneration was inhibited under this concentration range, revealing that this genotype presents a narrow phytohormonal range for shoot regeneration. Furthermore, higher BAP concentrations (>1 mg L^−1^) were essential for callus development on other *Bryophyllum* related species, avoiding organ formation [44].

Overall, ML tools allowed deciphering the concealed effects and interactions of phytohormones on *Bryophyllum* organogenesis. Future reports should focus on the analysis of somaclonal variation, especially on indirect shoots, to validate the suitability of this optimized protocol with the aim of obtaining true-to-type medicinal plants and by-products for their biotechnological exploitation.

## 5. Conclusions

In order to shed light on the concealed effects and interactions promoted by cytokinins and auxins on in vitro plant organogenesis, machine learning methodology emerges as a powerful tool, based on artificial intelligence technology. In our case, the use of neurofuzzy logic was able to predict and characterize the in vitro organogenesis of unexploited medicinal plants from the *Bryophyllum* subgenus. Our model identified the interaction between genotype and BAP concentration as the most significant factor driving organogenesis in these species. Furthermore, with the definition of model rules, we were able to make the interpretation of model results easier, by prioritizing the different ranges of BAP concentration and their differential effects on every species. Whereas BD and BH showed direct shoot regeneration in a wide range of BAP concentrations, from 0.375 to 1 mg L^−1^, BT showed a tighter range of BAP concentrations to ensure indirect shoot regeneration via callus formation, from 0.375 to 0.75 mg L^−1^. In addition, BT was the only species affected by IAA, which negatively influenced the development of indirect shoots. Consequently, the use of machine learning tools allowed deciphering the putative role of BAP on *Bryophyllum* in vitro organogenesis, showing a pleiotropic mechanism of action, depending on the genotype and concentration used.

To sum up, the combination of medicinal plant in vitro culture and machine learning technologies demonstrated that novel approaches should be applied to plant biotechnology to effectively respond to the current industrial requirements. In this case, the use of machine learning models revealed the mechanism of action of exogenous BAP and IAA, with the aim of optimizing an efficient plant tissue protocol, which could make the large-scale exploitation of *Bryophyllum* spp. easier.

## Figures and Tables

**Figure 1 biomolecules-10-00746-f001:**
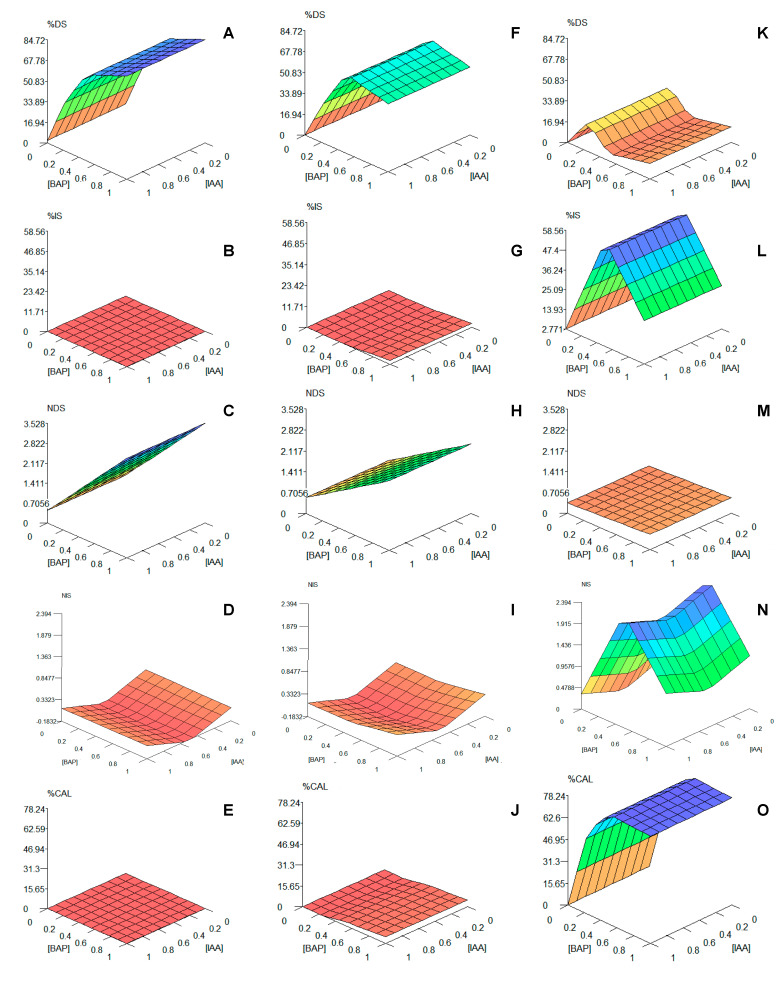
Graphical tridimensional extrapolation of the predictive models, generated for each output and genotype, by FormRules^®^: (**A**–**E**) results for *Bryophyllum* × *houghtonii* (BH); (**F**–**J**) results for *Bryophyllum daigremontianum* (BD); (**K**–**O**) results for *Bryophyllum tubiflorum* (BT); (**A**,**F**,**K**) refer to percentage direct shoot regeneration (%DS); (**B**,**G**,**L**) refer to percentage indirect shoot regeneration (%IS); (**C**,**H**,**M**) refer to number of direct shoots (NDS); (**D**,**I**,**N**) refer to number of indirect shoots (NIS); (**E**,**J**,**O**) refer to percentage of callus formation (%CAL). BAP and IAA concentrations are expressed as mg L^−1^.

**Figure 2 biomolecules-10-00746-f002:**
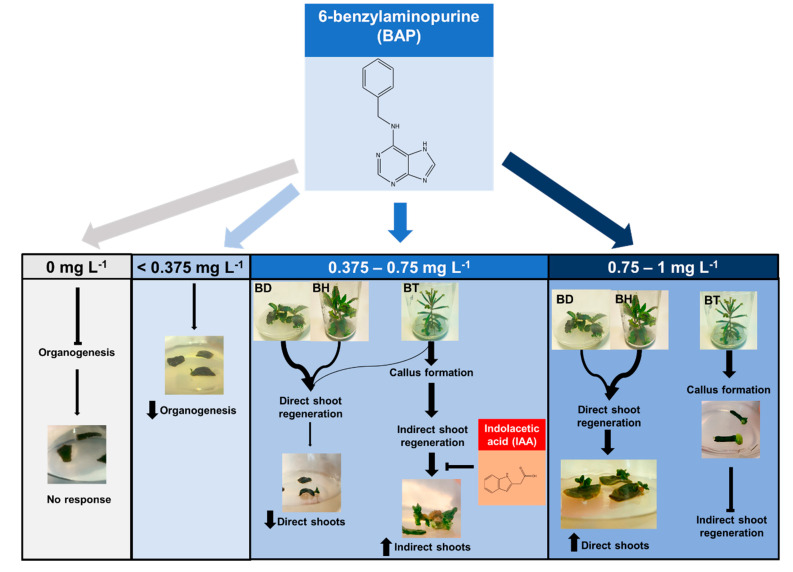
Proposed mechanism of action of the phytohormones BAP and IAA on the in vitro organogenesis of *Bryophyllum* spp., based on the predictive model generated by ML models.

**Figure 3 biomolecules-10-00746-f003:**
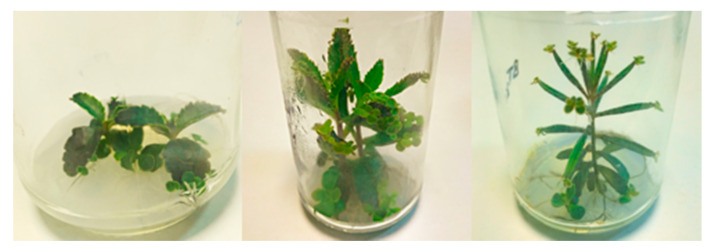
Foliar budding observed on donor plants of the *Bryophyllum* spp. BD (left) and BH (center) show foliar budding across the whole leaf margins. BT (right) shows restricted budding at the distal foliar end.

**Table 1 biomolecules-10-00746-t001:** Phytohormone treatments used for the *Bryophyllum* spp. organogenesis experiments. All treatments contained MS as the basal medium.

Treatment	(IAA) (mg L^−1^)	(BAP) (mg L^−1^)	Treatment	(IAA) (mg L^−1^)	(BAP) (mg L^−1^)
T01	0	0	T09	0.5	0
T02	0	0.25	T10	0.5	0.25
T03	0	0.5	T11	0.5	0.5
T04	0	1.0	T12	0.5	1.0
T05	0.25	0	T13	1.0	0
T06	0.25	0.25	T14	1.0	0.25
T07	0.25	0.5	T15	1.0	0.5
T08	0.25	1.0	T16	1.0	1.0

**Table 2 biomolecules-10-00746-t002:** Training parameters for model construction by FormRules 4.03^®^.

Minimization Parameters
MODEL SELECTION CRITERIA:
Cross Validation (CV)
Number of set densities: 2
Set densities: 2, 3
Adapt nodes: TRUE
Max. inputs per submodel: 4
Max. nodes per input: 15

**Table 3 biomolecules-10-00746-t003:** Results from organogenesis on *Bryophyllum* spp. cultured in vitro. The results for each output were expressed as the mean ± standard error (*n* = 12). Different letters indicate significant differences (*p* < 0.01). Bold values indicate the maximum value for each output.

Treat.	Genot.	IAA (mg L^−1^)	BAP (mg L^−1^)	%DS	%IS	NDS	NIS	%CAL	%DR
T01	BD	0	0	0.0 ± 0.0 ^d^	0.0 ± 0.0 ^d^	0.0 ± 0.0 ^d^	0.0 ± 0.0 ^d^	0.0 ± 0.0 ^d^	0.0 ± 0.0 ^a^
	BH	0	0	0.0 ± 0.0 ^d^	0.0 ± 0.0 ^d^	0.0 ± 0.0 ^d^	0.0 ± 0.0 ^d^	0.0 ± 0.0 ^d^	0.0 ± 0.0 ^a^
	BT	0	0	0.0 ± 0.0 ^d^	0.0 ± 0.0 ^d^	0.0 ± 0.0 ^d^	0.0 ± 0.0 ^d^	0.0 ± 0.0 ^d^	0.0 ± 0.0 ^a^
T02	BD	0	0.25	41.7 ± 7.0 ^bc^	0.0 ± 0.0 ^d^	2.3 ± 0.5 ^b^	0.0 ± 0.0 ^d^	0.0 ± 0.0 ^d^	0.0 ± 0.0 ^a^
	BH	0	0.25	61.1 ± 3.0 ^b^	0.0 ± 0.0 ^d^	2.0 ± 0.2 ^b^	0.0 ± 0.0 ^d^	0.0 ± 0.0 ^d^	0.0 ± 0.0 ^a^
	BT	0	0.25	27.8 ± 3.7 ^c^	50.0 ± 5.3 ^b^	1.7 ± 0.2 ^bc^	2.3 ± 0.4 ^bc^	72.2 ± 8.1 ^ab^	0.0 ± 0.0 ^a^
T03	BD	0	0.5	83.3 ± 9.1 ^ab^	0.0 ± 0.0 ^d^	1.9 ± 0.1 ^b^	0.0 ± 0.0 ^d^	0.0 ± 0.0 ^d^	0.0 ± 0.0 ^a^
	BH	0	0.5	52.8 ± 13.0 ^b^	0.0 ± 0.0 ^d^	2.0 ± 0.0 ^b^	0.0 ± 0.0 ^d^	0.0 ± 0.0 ^d^	11.1 ± 6.1 ^a^
	**BT**	**0**	**0.5**	11.1 ± 6.1 ^cd^	**88.9 ± 6.1 ^a^**	1.3 ± 0.7 ^bcd^	2.7 ± 0.1 ^b^	88.9 ± 6.1 ^a^	0.0 ± 0.0 ^a^
T04	BD	0	1.0	47.2 ± 1.9 ^bc^	0.0 ± 0.0 ^d^	1.8 ± 0.3 ^bc^	0.0 ± 0.0 ^d^	0.0 ± 0.0 ^d^	11.1 ± 6.1 ^a^
	**BH**	**0**	**1.0**	**100.0 ± 0.0 ^a^**	0.0 ± 0.0 ^d^	3.2 ± 0.2 ^ab^	0.0 ± 0.0 ^d^	0.0 ± 0.0 ^d^	0.0 ± 0.0 ^a^
	**BT**	**0**	**1.0**	0.0 ± 0.0 ^d^	44.4 ± 7.5 ^bc^	0.0 ± 0.0 ^d^	1.4 ±0.1 ^c^	**100.0 ± 0.0 ^a^**	0.0 ± 0.0 ^a^
T05	BD	0.25	0	0.0 ± 0.0 ^d^	0.0 ± 0.0 ^d^	0.0 ± 0.0 ^d^	0.0 ± 0.0 ^d^	0.0 ± 0.0 ^d^	0.0 ± 0.0 ^a^
	BH	0.25	0	0.0 ± 0.0 ^d^	0.0 ± 0.0 ^d^	0.0 ± 0.0 ^d^	0.0 ± 0.0 ^d^	0.0 ± 0.0 ^d^	11.1 ± 6.1 ^a^
	BT	0.25	0	0.0 ± 0.0 ^d^	0.0 ± 0.0 ^d^	0.0 ± 0.0 ^d^	0.0 ± 0.0 ^d^	0.0 ± 0.0 ^d^	0.0 ± 0.0 ^a^
T06	BD	0.25	0.25	19.4 ± 5.5 ^cd^	0.0 ± 0.0 ^d^	0.7 ± 0.2 ^cd^	0.0 ± 0.0 ^d^	0.0 ± 0.0 ^d^	8.3 ± 4.6 ^a^
	BH	0.25	0.25	41.7 ± 7.0 ^bc^	0.0 ± 0.0 ^d^	1.0 ± 0.0 ^c^	0.0 ± 0.0 ^d^	0.0 ± 0.0 ^d^	0.0 ± 0.0 ^a^
	**BT**	**0.25**	**0.25**	0.0 ± 0.0 ^d^	33.3 ± 10.5 ^c^	0.0 ± 0.0 ^d^	1.0 ± 0.5 ^cd^	**100.0 ± 0.0 ^a^**	0.0 ± 0.0 ^a^
T07	BD	0.25	0.5	33.3 ± 7.5 ^c^	0.0 ± 0.0 ^d^	1.8 ± 0.1 ^bc^	0.0 ± 0.0 ^d^	0.0 ± 0.0 ^d^	0.0 ± 0.0 ^a^
	BH	0.25	0.5	61.1 ± 3.0 ^b^	0.0 ± 0.0 ^d^	2.3 ± 0.2 ^b^	0.0 ± 0.0 ^d^	0.0 ± 0.0 ^d^	0.0 ± 0.0 ^a^
	**BT**	**0.25**	**0.5**	0.0 ± 0.0 ^d^	69.4 ± 10.7 ^ab^	0.0 ± 0.0 ^d^	3.1 ± 0.3 ^ab^	**100.0 ± 0.0 ^a^**	0.0 ± 0.0 ^a^
T08	**BD**	**0.25**	**1.0**	47.2 ± 1.9 ^bc^	0.0 ± 0.0 ^d^	2.2 ± 0.6 ^b^	0.0 ± 0.0 ^d^	0.0 ± 0.0 ^d^	**19.4 ± 1.9 ^a^**
	**BH**	**0.25**	**1.0**	88.9 ± 6.1 ^ab^	0.0 ± 0.0 ^d^	**3.7 ± 0.2 ^a^**	0.0 ± 0.0 ^d^	0.0 ± 0.0 ^d^	0.0 ± 0.0 ^a^
	BT	0.25	1.0	27.8 ± 3.7 ^c^	27.8 ± 3.7 ^c^	0.8 ± 0.1 ^cd^	1.3 ± 0.4 ^cd^	27.8 ± 3.7 ^c^	0.0 ± 0.0 ^a^
T09	BD	0.5	0	0.0 ± 0.0 ^d^	0.0 ± 0.0 ^d^	0.0 ± 0.0 ^d^	0.0 ± 0.0 ^d^	0.0 ± 0.0 ^d^	0.0 ± 0.0 ^a^
	BH	0.5	0	0.0 ± 0.0 ^d^	0.0 ± 0.0 ^d^	0.0 ± 0.0 ^d^	0.0 ± 0.0 ^d^	0.0 ± 0.0 ^d^	0.0 ± 0.0 ^a^
	BT	0.5	0	0.0 ± 0.0 ^d^	0.0 ± 0.0 ^d^	0.0 ± 0.0 ^d^	0.0 ± 0.0 ^d^	0.0 ± 0.0 ^d^	0.0 ± 0.0 ^a^
T10	BD	0.5	0.25	38.9 ± 3.7 ^bc^	0.0 ± 0.0 ^d^	1.2 ± 0.1 ^c^	0.0 ± 0.0 ^d^	0.0 ± 0.0 ^d^	0.0 ± 0.0 ^a^
	BH	0.5	0.25	38.9 ± 3.7 ^bc^	0.0 ± 0.0 ^d^	1.2 ± 0.1 ^c^	0.0 ± 0.0 ^d^	0.0 ± 0.0 ^d^	0.0 ± 0.0 ^a^
	BT	0.5	0.25	58.3 ± 5.6 ^b^	0.0 ± 0.0 ^d^	1.3 ± 0.1 ^c^	0.0 ± 0.0 ^d^	0.0 ± 0.0 ^d^	11.1 ± 6.1 ^a^
T11	BD	0.5	0.5	77.8 ± 6.1 ^ab^	0.0 ± 0.0 ^d^	2.3 ± 0.3 ^b^	0.0 ± 0.0 ^d^	0.0 ± 0.0 ^d^	0.0 ± 0.0 ^a^
	BH	0.5	0.5	88.9 ± 6.1 ^ab^	0.0 ± 0.0 ^d^	2.8 ± 0.1 ^ab^	0.0 ± 0.0 ^d^	0.0 ± 0.0 ^d^	0.0 ± 0.0 ^a^
	BT	0.5	0.5	0.0 ± 0.0 ^d^	55.6 ± 7.5 ^b^	0.0 ± 0.0 ^d^	1.8 ± 0.2 ^c^	88.9 ± 6.1 ^a^	0.0 ± 0.0 ^a^
T12	BD	0.5	1.0	47.2 ± 7.6 ^bc^	0.0 ± 0.0 ^d^	1.9 ± 0.3 ^b^	0.0 ± 0.0 ^d^	0.0 ± 0.0 ^d^	0.0 ± 0.0 ^a^
	BH	0.5	1.0	61.1 ± 3.0 ^b^	0.0 ± 0.0 ^d^	3.5 ± 0.3 ^ab^	0.0 ± 0.0 ^d^	0.0 ± 0.0 ^d^	0.0 ± 0.0 ^a^
	**BT**	**0.5**	**1.0**	0.0 ± 0.0 ^d^	8.3 ± 4.6 ^d^	0.0 ± 0.0 ^d^	0.3 ± 0.2 ^cd^	**100.0 ± 0.0 ^a^**	0.0 ± 0.0 ^a^
T13	BD	1.0	0	0.0 ± 0.0 ^d^	0.0 ± 0.0 ^d^	0.0 ± 0.0 ^d^	0.0 ± 0.0 ^d^	0.0 ± 0.0 ^d^	0.0 ± 0.0 ^a^
	BH	1.0	0	8.3 ± 4.6 ^d^	0.0 ± 0.0 ^d^	0.3 ± 0.2 ^cd^	0.0 ± 0.0 ^d^	0.0 ± 0.0 ^d^	16.7 ± 9.1 ^a^
	BT	1.0	0	0.0 ± 0.0 ^d^	0.0 ± 0.0 ^d^	0.0 ± 0.0 ^d^	0.0 ± 0.0 ^d^	0.0 ± 0.0 ^d^	0.0 ± 0.0 ^a^
T14	BD	1.0	0.25	44.4 ± 14.9 ^bc^	0.0 ± 0.0 ^d^	0.8 ± 0.1 ^cd^	0.0 ± 0.0 ^d^	0.0 ± 0.0 ^d^	0.0 ± 0.0 ^a^
	BH	1.0	0.25	36.1 ± 9.3 ^bc^	0.0 ± 0.0 ^d^	1.6 ± 0.1 ^bc^	0.0 ± 0.0 ^d^	0.0 ± 0.0 ^d^	0.0 ± 0.0 ^a^
	**BT**	**1.0**	**0.25**	8.3 ± 4.6 ^d^	69.4 ± 10.7 ^ab^	0.3 ± 0.2 ^cd^	**3.6 ± 0.7 ^a^**	69.4 ± 10.7 ^b^	0.0 ± 0.0 ^a^
T15	BD	1.0	0.5	61.1 ± 3.0 ^b^	0.0 ± 0.0 ^d^	2.5 ± 0.5 ^ab^	0.0 ± 0.0 ^d^	11.1 ± 6.1 ^cd^	0.0 ± 0.0 ^a^
	BH	1.0	0.5	80.6 ± 5.5 ^ab^	0.0 ± 0.0 ^d^	2.4 ± 0.2 ^ab^	0.0 ± 0.0 ^d^	0.0 ± 0.0 ^d^	0.0 ± 0.0 ^a^
	BT	1.0	0.5	11.1 ± 6.1 ^cd^	25.0 ± 3.7 ^c^	0.7 ± 0.4 ^cd^	1.2 ± 0.7 ^cd^	36.1 ± 7.5 ^c^	0.0 ± 0.0 ^a^
T16	BD	1.0	1.0	77.8 ± 6.1 ^ab^	8.3 ± 4.6 ^d^	1.8 ± 0.1 ^bc^	0.7 ± 0.4 ^cd^	19.4 ± 1.9 ^cd^	0.0 ± 0.0 ^a^
	BH	1.0	1.0	88.9 ± 6.1 ^ab^	0.0 ± 0.0 ^d^	2.7 ± 0.2 ^ab^	0.0 ± 0.0 ^d^	0.0 ± 0.0 ^d^	0.0 ± 0.0 ^a^
	BT	1.0	1.0	22.2 ± 6.1 ^c^	27.8 ± 3.7 ^c^	0.7 ± 0.2 ^cd^	1.2 ± 0.1 ^cd^	77.8 ± 6.1 ^ab^	0.0 ± 0.0 ^a^

**Table 4 biomolecules-10-00746-t004:** Critical factors and quality parameters detected by neurofuzzy logic. Bold inputs correspond to the strongest effect identified for every output.

Outputs	Train Set R^2^	*f* Ratio	df1, df2	*f* Critical (α > 0.05)	Significant Inputs
%DS	84.34	15.70	12, 47	1.97	**BAP** **× Genotype**
%IS	75.56	13.06	9, 47	2.09	**BAP** **× Genotype**
NDS	75.26	20.78	6, 47	2.30	**BAP** **× Genotype**
NIS	71.56	8.24	11, 47	2.00	**BAP** **× Genotype**
					IAA
%CAL	79.68	11.43	12, 47	1.97	**BAP** **× Genotype**
%DR	0.69	0.15	2, 47	3.20	-

**Table 5 biomolecules-10-00746-t005:** Rules generated by the model. The bolded rules indicate the inputs with the strongest effect on each output, with the highest and the lowest responses and their membership degree.

Rules		Genotype	BAP	IAA		%DS	%IS	NDS	NIS	%CAL	Membership
**1**	IF	**BD**	**LOW**		THEN	**LOW**					**1.00**
2		BD	MID LOW			LOW					0.64
3		BD	MID HIGH			HIGH					0.64
4		BD	HIGH			HIGH					0.55
5		BH	LOW			LOW					0.98
6		BH	MID LOW			LOW					0.56
7		BH	MID HIGH			HIGH					0.71
**8**		**BH**	**HIGH**			**HIGH**					**0.85**
**9**		**BT**	**LOW**			**LOW**					**1.00**
10		BT	MID LOW			LOW					0.76
11		BT	MID HIGH			LOW					0.94
12		BT	HIGH			LOW					0.88
**13**	IF	**BD**	**LOW**		THEN		**LOW**				**1.00**
**14**		**BD**	**MID**				**LOW**				**1.00**
15		BD	HIGH				LOW				0.98
**16**		**BH**	**LOW**				**LOW**				**1.00**
**17**		**BH**	**MID**				**LOW**				**1.00**
**18**		**BH**	**HIGH**				**LOW**				**1.00**
19		BT	LOW				LOW				0.97
**20**		**BT**	**MID**				**HIGH**				**0.70**
21		BT	HIGH				LOW				0.70
22	IF	BD	LOW		THEN			LOW			0.85
23		BD	HIGH					HIGH			0.63
24		BH	LOW					LOW			0.88
**25**		**BH**	**HIGH**					**HIGH**			**0.95**
**26**		**BT**	**LOW**					**LOW**			**0.90**
27		BT	HIGH					LOW			0.86
**28**	IF	**BD**	**LOW**		THEN				**LOW**		**1.00**
**29**		**BD**	**MID**						**LOW**		**1.00**
30		BD	HIGH						LOW		1.00
**31**		**BH**	**LOW**						**LOW**		**1.00**
**32**		**BH**	**MID**						**LOW**		**1.00**
**33**		**BH**	**HIGH**						**LOW**		**1.00**
34		BT	LOW						LOW		1.00
**35**		**BT**	**MID**						**HIGH**		**1.00**
36		BT	HIGH						LOW		0.59
37				LOW					LOW		0.75
38				MID					LOW		0.95
39				HIGH					LOW		0.75
**40**	IF	**BD**	**LOW**		THEN					**LOW**	**1.00**
**41**		**BD**	**MID LOW**							**LOW**	**1.00**
42		BD	MID HIGH							LOW	0.97
43		BD	HIGH							LOW	0.95
**44**		**BH**	**LOW**							**LOW**	**1.00**
**45**		**BH**	**MID LOW**							**LOW**	**1.00**
**46**		**BH**	**MID HIGH**							**LOW**	**1.00**
**47**		**BH**	**HIGH**							**LOW**	**1.00**
**48**		**BT**	**LOW**							**LOW**	**1.00**
49		BT	MID LOW							HIGH	0.60
**50**		**BT**	**MID HIGH**							**HIGH**	**0.78**
51		BT	HIGH							HIGH	0.76

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
