# Peer review of "Machine Learning Technology Reveals the Concealed Interactions of Phytohormones on Medicinal Plant In Vitro Organogenesis"

_biomolecules, 2020, doi:10.3390/biom10050746_

Round 1

Reviewer 1 Report

Dear Authors,

The idea of presented researches is really interesting, innovatory and of application importance so this article would interest many researchers from different fields. The experiments were designed well and the data support the discussion and conclusion.

I do not think any major modification is necessary for this paper and I recommend to publish this article in Biomolecules after correction of some minor suggestions:

  1. First of all, I think it is worth realizing that the reaction of plants observed in these experiments is a consequence of using exogenous phytohormones, in my opinion it should be emphasized to distinguished their influence from endogenous PGRs influence to avoid mistake with the endogenous PGR. From the results presented here we can only hypothesis the influence of exogenous IAA and BAP on the phytohormones endogenous level, transport phenomenon, etc.
  2. The disproportion between the extensive Discussion part and the relatively poorer Introduction is noticeable. That is why I suggest to transfer, for example, some information about the somaclonal variation in to the Introduction and in the Discussion section narrow down to specifics in this theme.
  3. I think it is better to place the Figure 2 and Figures 3 in the Result instead of Discussion section .
  4. Figure 3 needs the inscriptions and magnification – bar (but it is up to the Editor) and it should be indicate what it represents, it is not clear are they donor plants or regenerants?

And a few more particular suggestions:

  1. Line 16 “…which the role of plant hormones is usually underestimated” – better: crucial/important……
  2. Line 19 – insert: “in vitro” to achieve: “in vitro organogenesis”
  3. Line 41 – insert: “in vitro” to achieve: “In vitro organogenesis”
  4. Line 112 – the full name of IAA and BAP here is not necessary
  5. Line 436 - insert: “in vitro” to achieve: “in vitro organogenesis”

Author Response

Dear Referee:

We truly acknowledge the time and effort used to send these comments on your review.

Please, kindly see our comments, in attached file, which will be inserted in blue among your comments (in black).

Dear Authors,

The idea of presented researches is really interesting, innovatory and of application importance so this article would interest many researchers from different fields. The experiments were designed well and the data support the discussion and conclusion.

I do not think any major modification is necessary for this paper and I recommend to publish this article in Biomolecules after correction of some minor suggestions:

First of all, I think it is worth realizing that the reaction of plants observed in these experiments is a consequence of using exogenous phytohormones, in my opinion it should be emphasized to distinguished their influence from endogenous PGRs influence to avoid mistake with the endogenous PGR. From the results presented here we can only hypothesis the influence of exogenous IAA and BAP on the phytohormones endogenous level, transport phenomenon, etc.

We have reinforced the exogenous nature of the applied phytohormones. Please, see lines 60, 70, 98, 106 and 325-26.

The disproportion between the extensive Discussion part and the relatively poorer Introduction is noticeable. That is why I suggest to transfer, for example, some information about the somaclonal variation in to the Introduction and in the Discussion section narrow down to specifics in this theme.

We have balanced the extension between the introduction and the discussion. See lines 64-72.

I think it is better to place the Figure 2 and Figures 3 in the Result instead of Discussion section.

As recommended, we have placed Figure 2 at  the Results sections.

In our opinion it is better to keep Figure 3 at the Discussion section because it aids to understand what is stated during discussion, reinforcing our viewpoints and hypothesis about foliar budding control by endogenous phytohormones. However, if we are required to modify it, we will.

Figure 3 needs the inscriptions and magnification – bar (but it is up to the Editor) and it should be indicate what it represents, it is not clear are they donor plants or regenerants?

We send the figure in jpg format. If further improvements in image quality are necessary, please  let us know.

Figure 3 represents donor plants: it is now included, see line 413. Thank you.

And a few more particular suggestions:

Line 16 “…which the role of plant hormones is usually underestimated” – better: crucial/important……

Done.

Line 19 – insert: “in vitro” to achieve: “in vitro organogenesis”

Done. See line 18.

Line 41 – insert: “in vitro” to achieve: “In vitro organogenesis”

Done. Also we have added in vitro organogenesis thoroughly in the manuscript.

Line 112 – the full name of IAA and BAP here is not necessary

Done.

Line 436 - insert: “in vitro” to achieve: “in vitro organogenesis”

Done.

Reviewer 2 Report

The ‘biomolecules-796661’ manuscript addresses the utilization of neurofuzzy logic - a combination of artificial neural networks (ANNs) and fuzzy logic algorithms - in plant biotechnology, to characterize the mechanism of action of two phytohormones, leading to the optimization of plant tissue culture protocols.

However the work is interesting and labor-intensive as well as the methods are acceptable, there are problems with the presentation of the manuscript. Fig.2., which is a very important figure to present the results of the work using ANN software tool, has poor quality and there are problems with the content too. Please, check it. Use the correct scientific sign (T-end line) of the inhibitor effect on the figure instead of the strange and slurred “no entrance” traffic sign.

Author Response

Dear Referee,

We truly acknowledge your comments that will surely improve the overall quality of the manuscript. Our comments are inserted in blue in attached file.

The ‘biomolecules-796661’ manuscript addresses the utilization of neurofuzzy logic - a combination of artificial neural networks (ANNs) and fuzzy logic algorithms - in plant biotechnology, to characterize the mechanism of action of two phytohormones, leading to the optimization of plant tissue culture protocols.

However the work is interesting and labor-intensive as well as the methods are acceptable, there are problems with the presentation of the manuscript. Fig.2., which is a very important figure to present the results of the work using ANN software tool, has poor quality and there are problems with the content too. Please, check it. Use the correct scientific sign (T-end line) of the inhibitor effect on the figure instead of the strange and slurred “no entrance” traffic sign.

Done. We  have changed the inhibition arrows and improved the quality of the figure 2.

Reviewer 3 Report

The paper entitled “Machine learning technology reveals the concealed interactions of phytohormones on medicinal plant in vitro organogenesis”, describes the use of a powerful tool for data modeling —named machine learning (ML) technology—for  revealing the critical factors that affect a multifactorial organogenesis process. For this, authors used 3 genotypes of Bryophyllum subgenus, 2 hormones (AIA and BAP) at 4 different concentrations (ranging from 0 to 1 mg/l) –which make a total of 48 combinations of independent variables or inputs — for developing a regeneration experiment in which 6 organogenesis parameters (dependent variables or outputs) were analyzed. A complex set of data was analyzed by using ML technology which allows us to better understand the relationship among the variables in order to establish an optimized regeneration protocol. The modeling graphic allows an easy interpretation of the results, and the relationships among variables are visible. The results and conclusions are clear and well explained. Although the bioinformatic part could be difficult to understand for non-experts, the authors explain the procedures and the results obtained very well, and the paper is written correctly.

Nevertheless, these observations should be considered:

The introduction provides sufficient information, but from my point of view, the objective is not totally clear (line 87 to 91): “our study is committed to provide insight about the critical factors that influence organogenesis in order to achieve the valorization of true-to-type products from...)”

However, I cannot see if the authors made any experiment for valorizing  “true-to-type products”. The aim of this work must be rewritten with the real goal as: “our study is committed to providing insight into the critical factors that influence organogenesis of Bryophyllum spp. cultured in vitro, by focusing on the effects developed by phytohormones: the auxin indoleacetic acid (IAA) and the cytokinin 6-benzylaminopurine (BAP)”.

Material and methods are well described even when the experiment is complex, but some explanations should be made about the experimental procedure:

The plant multiplication specifications confuse me:

Line 102-103: “Plant multiplication was carried out every 12 weeks by using newly-formed epiphyllous buds as the propagation explant” but then in Line 105-106 the authors state: “Foliar disks (≅ 1 cm2) from 12-week-old in vitro-grown plants (6th subculture) were excised and used for the subsequent organogenesis experiments”. And then in lines 113-115: “Three leaf disks were placed into each culture vessel and four vessels were used for each treatment. All vessels were placed randomly in a growth chamber under the same conditions described above. After 8 weeks, six parameters were determined to analyze organogenesis …”

Regarding this, I don’t understand: How many subcultures did you make? Which material did you use for counting the indirect or direct organogenesis? Did you count the first new-bud formed under the initial explants? After how many weeks?

Which is the number of biological replicates that you consider for each treatment? n=4, n=12… This should be explained in the procedure section and also indicated in table captions: mean + standard error (n=??)

Some paragraphs can be considered as an introduction such as those in lines 125 to 128 and 144 to 148, or results, such as those  comments in between 139 to 142 lines.

Statistical analysis should be supplied at least in supplementary data.

Results

Results are clearly presented but I have some observations about that:

Table 3: the number of replicates must be included.

Line 204-205: I think that the authors actually wanted to say: “indirect rooting was not detected in this study..”, isn't it?

Table 4: How do you establish if an input has a strong effect or not? Because the input is significant? In this sense, AIA effect on NIS variable is significant but not strong? I don´t understand  the terminology well.

Line 226-228: The fact that there are not significant differences between experimental and model-predictive data by ANOVA is good, isn't it? Authors should clarify this.

Table 5: What do the letters (I, F, T, H…) mean in the table? it is a statistical analysis? This must be indicated in the caption and explained in the text.

Regarding the value “membership”, what does it mean? An explanation in the text could help in the same way as the authors explain the levels of phytohormones.

Figure 1: Please, increase the size of the numbers on both axes and the names of the variables because they are very small.

Line 284-286: In this final paragraph, maybe the authors could make reference to the ML technology more than to the ANN software.

 Discussion

Line 307: cite 26. I am not sure that this reference will be ok for this sentence

Line 312: include in the sentence… “adequate phytohormonal balance for each genotype in order to…”

The focus of the discussion must be revised. Have authors  got previous data about the genetic variation frequency for this species? or  Do you know if it is difficult or not to produce somaclonal variation in Bryophyllum by in vitro culture?

If you don't have information about that, I think it is speculative to relate your data to somaclonal variation. I propose to change the discussion a little bit because it is very focused on somaclonal variation when the authors haven't got enough information about the relation between direct or indirect organogenesis and somaclonal variation. Also, at the end of the discussion, the authors recognize, in lines 420-422: “Future reports should focus on the analysis of somaclonal variation, especially on indirect shoots, to validate the suitability of this optimized protocol with the aim of obtaining true-to-type medicinal plants and by-products for their biotechnological exploitation.”

Line 330: The word “missing” confuses me. Maybe authors wanted to say: “The only parameter that was not predictable after modeling was %DR”.

Figure 2: I propose to use Figure 2 as Abstract Figure. I think it is very illustrative.

Line 382-383: include “data not shown”

Figure 3: I think that discussion is not the proper section for including new figures or results. I suggest moving Fig 3 to a results or supplementary section. Moreover, these results are not described in material and methods section.

 Line 401: include: “…this genotype probably contains..”

Conclusions

The conclusions are clear and well supported by the results.

Line 440-442: Again in this final paragraph, authors must make reference to the ML technology more than to the ANN software or change the title of the work including ANN software in it.

Supplementary materials

Please, increase the size of the numbers on both axes and the names of the variables because they are very small.

Author Response

Dear Referee,

We really appreciate and acknowledge the comments made in the course of the reviewing process of our manuscript. Please, kindly see all our responses (in blue) to your comments (black), that surely will improve the quality of the manuscript.

The paper entitled “Machine learning technology reveals the concealed interactions of phytohormones on medicinal plant in vitro organogenesis”, describes the use of a powerful tool for data modeling —named machine learning (ML) technology—for  revealing the critical factors that affect a multifactorial organogenesis process. For this, authors used 3 genotypes of Bryophyllum subgenus, 2 hormones (AIA and BAP) at 4 different concentrations (ranging from 0 to 1 mg/l) –which make a total of 48 combinations of independent variables or inputs — for developing a regeneration experiment in which 6 organogenesis parameters (dependent variables or outputs) were analyzed. A complex set of data was analyzed by using ML technology which allows us to better understand the relationship among the variables in order to establish an optimized regeneration protocol. The modeling graphic allows an easy interpretation of the results, and the relationships among variables are visible. The results and conclusions are clear and well explained. Although the bioinformatic part could be difficult to understand for non-experts, the authors explain the procedures and the results obtained very well, and the paper is written correctly.

Nevertheless, these observations should be considered:

The introduction provides sufficient information, but from my point of view, the objective is not totally clear (line 87 to 91): “our study is committed to provide insight about the critical factors that influence organogenesis in order to achieve the valorization of true-to-type products from...)” However, I cannot see if the authors made any experiment for valorizing  “true-to-type products”. The aim of this work must be rewritten with the real goal as: “our study is committed to providing insight into the critical factors that influence organogenesis of Bryophyllum spp. cultured in vitro, by focusing on the effects developed by phytohormones: the auxin indoleacetic acid (IAA) and the cytokinin 6-benzylaminopurine (BAP)”.

Done. The reviewer is right: please find a more concise objective, see lines 104-107.

Material and methods are well described even when the experiment is complex, but some explanations should be made about the experimental procedure:

The plant multiplication specifications confuse me:

Line 102-103: “Plant multiplication was carried out every 12 weeks by using newly-formed epiphyllous buds as the propagation explant” but then in Line 105-106 the authors state: “Foliar disks (≅ 1 cm2) from 12-week-old in vitro-grown plants (6th subculture) were excised and used for the subsequent organogenesis experiments”. And then in lines 113-115: “Three leaf disks were placed into each culture vessel and four vessels were used for each treatment. All vessels were placed randomly in a growth chamber under the same conditions described above. After 8 weeks, six parameters were determined to analyze organogenesis …”. Regarding this, I don’t understand: How many subcultures did you make? Which material did you use for counting the indirect or direct organogenesis? Did you count the first new-bud formed under the initial explants? After how many weeks?

We have improved this section. See lines 118-126. Briefly:

After disinfection, in vitro-established plantlets were subcultured every 12 weeks to fresh media and only the plants from the 6 subculture (72 weeks) were used to get the foliar disks used in the in vitro organogenesis experiments. Then, three foliar disks were placed on each culture vessel, and four culture vessels were used for every treatment.  Then the number of sampled was 12. After 8 weeks, the organogenesis parameters were evaluated from the responses observed on these foliar disks.

Which is the number of biological replicates that you consider for each treatment? n=4, n=12… This should be explained in the procedure section and also indicated in table captions: mean + standard error (n=??)

Twelve biological replicates were considered for each treatment, n=12. We have included this information in lines 125 and 190.

Some paragraphs can be considered as an introduction such as those in lines 125 to 128 and 144 to 148, or results, such as those comments in between 139 to 142 lines.

Done. We have reorganized these paragraphs: see lines 79-81 and 90-93.

Results

Results are clearly presented but I have some observations about that:

Table 3: the number of replicates must be included.

Done. See line 190.

Line 204-205: I think that the authors actually wanted to say: “indirect rooting was not detected in this study..”, isn't it?

Done. See lines 226.

Table 4: How do you establish if an input has a strong effect or not? Because the input is significant? In this sense, AIA effect on NIS variable is significant but not strong? I don´t understand  the terminology well.

Unlike ANN models, which lead to “black box mathematical models”, neurofuzzy logic tools allow you to build "gray mathematical models" because they are capable of expressing the outputs from the mathematical model using a set of IF-THEN linguistic rules. These rules help to generate knowledge about the studied process, but require some interpretation. The software that we have used, FormRules, divides the model into several submodels to develop simpler rules. In this case, each submodel includes 1 or 2 inputs that explain the variations of the output in the database. The submodel that most affects a particular output is highlighted. The IF-THEN rule that indicates the "strongest or greatest positive effect" is colored in blue and the one that indicates the greatest negative effect is colored in red.

As an example in Table 4, for the output NIS, the variations in the three inputs BAP, Genotype and IAA contribute to explain the variations in the output NIS, but the effect of BAP and Genotype is stronger than the effect of IAA.

Line 226-228: The fact that there are not significant differences between experimental and model-predictive data by ANOVA is good, isn't it? Authors should clarify this. Statistical analysis should be supplied at least in supplementary data.

Referee is right. Yes, if the calculated f ratio is higher than f critical (Table 4), means that the model predict accurately. Not statistically differences exist between the experimental data and the predicted by the model. In this sense, we indicate “the predictability of the model was assessed”.

Table 5: What do the letters (I, F, T, H…) mean in the table? it is a statistical analysis? This must be indicated in the caption and explained in the text.

Sorry. It was just the table format than change the meaning. Now you can see the real meaning as “IF-THEN” rules. We expressed it clearer, see Table 5.

Regarding the value “membership”, what does it mean? An explanation in the text could help in the same way as the authors explain the levels of phytohormones.

A new paragraph explaining the meaning of membership degree (supported further detailed explanations in our previous papers) has been added. See lines 166-167)

Figure 1: Please, increase the size of the numbers on both axes and the names of the variables because they are very small.

Done. We have enlarged them as much as possible. We will send a new figure in jpg format to the editor to improve the quality.

Line 284-286: In this final paragraph, maybe the authors could make reference to the ML technology more than to the ANN software.

Done. See Line 252.

 Discussion

Line 307: cite 26. I am not sure that this reference will be ok for this sentence

Done. We have replaced the reference, see new reference 12 in line 70.

Line 312: include in the sentence… “adequate phytohormonal balance for each genotype in order to…”

Done. See lines 71 and 72.

The focus of the discussion must be revised. Have authors  got previous data about the genetic variation frequency for this species? or  Do you know if it is difficult or not to produce somaclonal variation in Bryophyllum by in vitro culture? If you don't have information about that, I think it is speculative to relate your data to somaclonal variation. I propose to change the discussion a little bit because it is very focused on somaclonal variation when the authors haven't got enough information about the relation between direct or indirect organogenesis and somaclonal variation. Also, at the end of the discussion, the authors recognize, in lines 420-422: “Future reports should focus on the analysis of somaclonal variation, especially on indirect shoots, to validate the suitability of this optimized protocol with the aim of obtaining true-to-type medicinal plants and by-products for their biotechnological exploitation.”

The reviewer is right. We modified the manuscript because of this concern. Somaclonal information was relocated in the introduction as a side factor related to indirect organogenesis. See lines 64-72.

Line 330: The word “missing” confuses me. Maybe authors wanted to say: “The only parameter that was not predictable after modeling was %DR”.

Done. See line 343.

Figure 2: I propose to use Figure 2 as Abstract Figure. I think it is very illustrative.

We will suggest it to the Editor.

Line 382-383: include “data not shown”

Done. See line 387.

Figure 3: I think that discussion is not the proper section for including new figures or results. I suggest moving Fig 3 to a results or supplementary section. Moreover, these results are not described in material and methods section.

We only included this Figure as a support to what is described and found in the literature, it was not a result but to illustrate each species.

 Line 401: include: “…this genotype probably contains..”

Done. See line 406.

Conclusions

The conclusions are clear and well supported by the results.

Line 440-442: Again in this final paragraph, authors must make reference to the ML technology more than to the ANN software or change the title of the work including ANN software in it.

Done. See line 445-446.

Supplementary materials

Please, increase the size of the numbers on both axes and the names of the variables because they are very small.

Done. See new Figure S1. Also, we will submit another image file directly to the Editor to improve image quality.